# Outcome Predictors and Safety of Home Dobutamine Intravenous Infusion in End Stage Heart Failure Patients

**DOI:** 10.3390/jcm10122571

**Published:** 2021-06-10

**Authors:** Antoine Jobbé-Duval, Thomas Bochaton, Guillaume Baudry, Eric Bonnefoy-Cudraz, Elisabeth Hugon-Vallet, Matteo Pozzi, Jean-Francois Obadia, Danka Tomasevic, Camille Amaz, Nathan Mewton, Laurent Sebbag

**Affiliations:** 1Heart Failure and Transplant Department, Hospices Civils de Lyon, “Louis Pradel” Cardiologic Hospital, 69002 Lyon, France; guillaume.baudry@chu-lyon.fr (G.B.); elisabeth.hugon-vallet@chu-lyon.fr (E.H.-V.); nathan.mewton@chu-lyon.fr (N.M.); laurent.sebbag@chu-lyon.fr (L.S.); 2Cardiac Intensive Care Unit, Hospices Civils de Lyon, “Louis Pradel” Cardiologic Hospital, 69002 Lyon, France; thomas.bochaton@chu-lyon.fr (T.B.); eric.bonnefoy-cudraz@chu-lyon.fr (E.B.-C.); danka.tomasevic@chu-lyon.fr (D.T.); 3Department of Cardiac Surgery, Hospices Civils de Lyon, “Louis Pradel” Cardiologic Hospital, 69002 Lyon, France; matteo.pozzi@chu-lyon.fr (M.P.); jean-francois.obadia@chu-lyon.fr (J.-F.O.); 4Clinical Investigation Center, Hospices Civils de Lyon, “Louis Pradel” Cardiologic Hospital, 69002 Lyon, France; camille.amaz@chu-lyon.fr

**Keywords:** end-stage heart failure, home dobutamine, mortality, safety

## Abstract

Patients in end-stage heart failure can experiment cardiogenic shock and may not be weanable from dobutamine. The fate of these patients is a challenge for doctors, patients, family, and the institution. Dobutamine use at home can be a solution. The aim of the present study was to assess the outcome, biological predictors, and safety of dobutamine use at home in dobutamine-dependent patients. All consecutive dobutamine-dependent patients discharged with continuous home intravenous dobutamine, from a single tertiary center between February 2014 and November 2019, were retrospectively analyzed. A total of 19 patients (age 65 ± 10 years) were followed for one year. At one-year, the survival rate was 32%, (6/19). Five (26%) patients had an adverse event related to the intravenous catheter. In a multivariate logistic regression analysis, the combination of a glomerular filtration rate >60 mL/min and a brain natriuretic peptide level <1000 ng/L, were highly predictive of one-year survival (HR = 10.87, IC95% (5.78–36.44), *p* < 0.001). Management of dobutamine-unweanable patients after cardiogenic shock may involve dobutamine at home to permit a home return. This strategy allows a significant survival and few readmissions, and, if eligible, access to surgical strategies, such as heart transplantation. Simple biological markers at discharge can identify severe patients to refer to palliative care and good responders.

## 1. Introduction

Despite all the progress made in heart failure therapies, up to 10% of heart failure patients ultimately progress towards advanced and end-stage heart failure. At this stage, patients are symptomatic, refractory to medical treatment optimization according to guidelines and have a poor quality of life [1]. Therapeutic options for these severe patients are limited to short and long-term mechanical cardiac support or heart transplantation. Most patients do not benefit from these therapies due to a lack of grafts or their Left Ventricular Assist Device (LVAD) ineligibility. Cardiogenic shock is one of the final courses of the disease and justifies inotropic treatments to maintain a hemodynamic stability. The most severe patients are not weanable from cardiac inotropes. These patients are stable but cannot be discharged from the hospital. They are classified by the Intermacs scale at the Intermacs 3 stage [2], stable under inotropic drugs. The outcome of these patients is challenging, as they are stabilized by the inotropic treatment but stranded in hospital by the continuous central venous administration of the treatment accompanied by daily nursing supervision.

The treatment strategy can then be two-fold, pending a heart transplant or a patient decision for a cardiac surgery such as Mitraclip or LVAD (bridge to therapy: BTTh), or palliative (destination therapy: DT).

In these patients, intravenous home dobutamine infusion can be achieved under daily nursing supervision via home hospitalization. This is not a routine procedure and there are few data on this type of management in the literature [3,4,5,6,7]. In the REMATCH study [8], the medical group included 72% of patients that were not weanable from intravenous inotropic treatment, and this group had a high mortality rate of 75% at two years of follow-up. There are also few data on factors associated with mortality in this patient population.

The objectives of our study were to describe the characteristics and the safety of intravenous home dobutamine infusion and assess the factors associated with survival.

## 2. Materials and Methods

### 2.1. Study Population

This study is a retrospective study of all patients with advanced or end-stage heart failure (Intermacs 3 Stage) discharged from a single tertiary referral center with intravenous home dobutamine (IHD). All patients were admitted for cardiogenic shock and were not weanable from dobutamine. The study protocol was reviewed and approved by our local ethics committee and a waiver of consent was given by our administrative authorities.

All patients were hospitalized in our critical care unit for cardiogenic shock, and were placed on continuous dobutamine infusion to maintain a cardiac index > 2.0 L/min/m^2^, and/or a mean arterial pressure >65 mmHg and/or a daily diuresis >500 mL/day.

In this registry, IHD was applied to two different groups of end-stage heart failure patients: the bridge to therapy (BTTh) group, where IHD was placed transiently pending a surgical intervention or an advanced therapy (percutaneous mitral valve repair, left ventricular assist device, heart transplant) or the destination therapy (DT) group, in patients not eligible for advanced therapy, in a palliative setting.

All cases were reviewed by the local heart team, comprising an advanced heart failure and heart transplant specialist, cardiac critical care specialist, intensive care specialist and heart surgeon.

### 2.2. Ambulatory Treatment Protocol

The dobutamine infusion rate was set at the discretion of the attending senior physician in the critical cardiac care unit to maintain the lowest dose of dobutamine allowing a mean arterial pressure >65 mmHg and a daily diuresis ≥500 mL/day. Then, when all patients were stabilized and hospital discharge was decided, all the patients received a single-lumen peripheral intravenous central catheter (PICC) line to allow continuous intravenous home dobutamine. Beta-blockers were stopped during the hospitalization.

For home administration, patients received an automated continuous infusion pump. These patients were referred to a home hospitalization team with a home nurse refilling and changing the dobutamine syringes daily with aseptic bandages of the PICC line. The ambulatory dosing was set to be the same as the one administered in hospital. Blood pressure, heart rate, diuresis, and oxygen saturation were monitored twice daily by this home nurse. A coordinating doctor visited once every week.

### 2.3. Data Collection and Follow-Up

All data from this population were collected retrospectively from their electronic medical records and all other available medical records. After initiation of the therapy, patients remained in regular follow-up at the outpatient clinic with a follow-up schedule that was left at the discretion of the treating physician, usually between 4–12 weeks. All adverse events during follow-up were carefully collected from electronic medical records or direct medical contact. Adverse events collected included:Death from any cause;Heart transplantation or left ventricular assist device implantation;Dobutamine withdrawal;Cardiac surgery;Number of days spent hospitalized in cardiology whatever the cause;Intravenous line adverse events: infection and endocarditis, hematoma, displacement/section, thrombosis, replacement.

### 2.4. Statistics

Continuous variables are expressed as mean ± standard deviation or median (interquartile range) and were compared between groups with the Student’s *t*-test of Mann–Whitney U test as appropriate (depending on normal distribution). Categorical data were expressed as number and percentages and compared with the Chi-square test or Fisher exact test, as appropriate.

We compared patients who died during follow-up to patients alive at one-year on the different parameters available via an associated non-parametric test (Wilcoxon-Mann-Whitney). Follow-up was performed in each patient up to one-year after IHD start and all death events were recorded, as well as pre-specified adverse events. Univariate regressions analyzes and multivariate cox regression models were built to assess the relationship of each baseline characteristic with all-cause death at one-year. All variables that had a significant relationship with all-cause death in the univariate Cox regression, were integrated in the multivariate regression analysis. We looked for thresholds of glomerular filtration rate (GFR) and brain natriuretic peptide (BNP) levels to predict death at one-year using receiver operating characteristic curves. Based on these analyses, the median glomerular filtration rate and the median BNP levels in our cohorts appeared to have a strong predictive relationship with death at one-year. We therefore built a simple multivariate Cox regression model with these two variables (BNP > 1000 ng/L; GFR < 60 mL/min/m^2^). We then constructed a receiver operating characteristic curve for BNP alone, GFR alone and combination of GFR and BNP and we compared the area under the curve of each model with the C-statistic. We also constructed the 1-year survival curves according to median BNP and GFR values and their combination (one high-risk group with BNP > 1000 ng/L and GFR < 60 mL/min/m^2^ versus all others). Kaplan–Meier curves were compared using the Log rank test.

No adjustment for multiple testing was performed, as all of our analyses can be considered as exploratory. Statistical significance was set at a two-tailed probability level of <0.05.

Statistics were performed using R (version 3.3.3, R Foundation for Statistical Computing, Vienna, Austria).

## 3. Results

### 3.1. Characteristics of the Population

We included 19 end-stage heart failure patients at our hospital from February 2014 to November 2019. During this study period, 216 adult patients were listed for heart transplant, 198 patients had a heart transplant and 62 patients had an LVAD implantation (of which 42 were BTTh and 20 were DT).

Table 1 shows the baseline characteristics of the patients at the admission for cardiogenic shock.

Twelve patients (63%) had IHD in a palliative setting and seven (37%) in a bridge-to-treatment setting. All BTTh patients had a contraindication (RV failure) or refused LVAD surgery. One patient finally accepted LVAD implantation.

Patients were 65 ± 10 years old, with LVEF 21 ± 5% and two-thirds had an ischemic cardiomyopathy. The mean dose of dobutamine at discharge was 2.6 ± 1.2 µg/kg/min. The baseline biomarkers for the patient population, as well as the hospital discharge and last follow-up results are reported in Table 2.

There was a significant improvement in creatinine and glomerular filtration rate values at follow-up during IHD. There was no significant improvement in BNP levels at follow-up.

### 3.2. Clinical Outcomes and Mortality at One-Year

The median follow-up for our patient population was 203 (79–434) days. There was no loss of sight during the follow-up. There were no changes in intravenous dobutamine rates during the home hospitalization period. The clinical outcomes at one-year follow-up are reported in detail in Table 3.

During the intravenous dobutamine treatment period (121 (40–299) days), our population was out of hospital for 116 (39–265) days, the total number of rehospitalization days was 31 (12–45) days.

At one year, a total of six patients still under dobutamine therapy (32%) survived, three in the group DT (25%), three in the group BTTh (43%). The Kaplan–Meier survival curve for the whole study population is presented in Figure 1.

In the 7 patients included in the BTTh group, 5 (71%) patients finally underwent surgical treatment: 3 (42%) underwent heart transplantation, 1 (14%) had a percutaneous mitral valve repair, and one (14%) had a left ventricular assistance device implantation.

In the 12 patients included in the DT group, 2 patients (17%) were weaned late from dobutamine without the need to reintroduce treatment.

Of the 13 patients who died, 11 patients died while on dobutamine infusion. The causes of death were progressive pump failure in six patients, severe endocarditis with cardiogenic shock or sepsis in five patients, and two patients died of acute graft failure after heart transplantation.

### 3.3. Safety

During the entire follow-up, a total of five (26%) PICC lines had to be removed for infection, with four cases of endocarditis of the tricuspid valve or pace-maker endocarditis. These endocarditis cases were bacterial infections (*Staphylococcus aureus*) in four cases or fungus infection (*Candida albicans*) in one patient. Four of these cases were in DT patients and one in a BTTh patient and were all complicated by death.

There were no reported cases of PICC line thrombosis, malfunction, migration, or occlusion in our patient population. No one required reoperation. There were also no reported cases of hypereosinophilia despite the duration of dobutamine infusion.

### 3.4. Predictors of Mortality at One-Year

The clinical and biological variables that were associated with one-year mortality in univariate and multivariate analyses are reported in Table 4.

The only variable that was independently predictive of all-cause death at one-year were a GFR < 60 mL/min (HR = 7.04 95%CI (1.32; 37.48); *p* = 0.02).

The event-free survival curves combining GFR and BNP thresholds to define groups are presented in Figure 2.

Based upon the multivariate analysis findings, we constructed a predictive model integrating GFR and BNP to predict death events within one year. Receiver operating characteristic curves were built and the integration of these two markers yielded an area under the curve of 95%CI (0.85; 0.99). The corresponding ROC curve is presented in Figure 3.

## 4. Discussion

Home dobutamine permits the home return of severe advanced heart failure patients unweanable from dobutamine therapy; it allows extended home stay for patients, and probably fewer rehospitalizations. However, in our study, it was associated with a significant incidence of catheter infections.

### 4.1. Home Return and Waiting Strategy

Patients with cardiogenic shock that cannot be weaned from dobutamine are clinically difficult, given the contraindication to LVAD and the delay in obtaining a transplant. Home inotropes can be an option during the waiting time for heart transplant or to permit a home discharge from the intensive cardiology unit [4,9].

In our population, patients stayed at home the most part of the dobutamine home treatment (200 days) with very few rehospitalization (30 days). In our BTTh population (seven patients), five patients could access a surgery procedure (three heart transplantation, one LVAD, one Mitraclip). Waiting strategy on home inotrope has already been evaluated in larger populations with encouraging results [9]. Milrinone seems to provide an interesting option when the waiting time before heart transplantation is short (<100 days). A longer waiting time should lead to consideration of a LVAD strategy, if right ventricular function is compatible with LVAD implantation.

### 4.2. Survival Results on Home Dobutamine

The survival of these Intermacs 3 patients is compromised in the very short term due to the severity of their cardiac condition. The maintenance of therapy via continuous administration by a Picc line and the support of home hospitalization has enabled a significant increase in their survival. We did not have a control group as stopping treatment in these patients being associated with rapid cardiac death.

This therapy achieves a survival of 32% at one-year, corresponding to the survival of the patients described in the REMATCH study [8] (LVAD versus medical treatment) where the control population was on dobutamine in 72% of cases.

We highlight two criteria associated with mortality in our population, renal function and the rate of natriuretic peptides under dobutamine. Creatinine clearance has statistically significant value, but not BNP, due to a lack of potency in a small population.

We found that markers were significantly associated with mortality in our population: renal function and the natriuretic peptides levels under dobutamine. The association of both biomarkers (GFR < 60 mL/min and BNP > 1000 ng/L) at hospital discharge identifies an extremely severe patient population with high mortality (0% survival at six months). Patients discharged from the hospital with GFR > 60 mL/min and BNP < 1000 ng/L are those who seem to have an excellent survival with dobutamine treatment at home (100% survival at six months).

B-type natriuretic peptide [10,11,12] and creatinine levels [13,14] have already been described as strong prognosis biomarkers in the general heart failure population, but not in this advanced heart failure population.

### 4.3. Strategy Complications

Twenty-five percent of the study population had catheter infections. They were all associated with a fatal outcome. This infectious complication rate is the same as described in the literature, between 15 and 34% considering the reports [4,9].

We did not observe any signs of inflammatory reaction to dobutamine over the entire duration of the follow-up. Furthermore, we did not find any induced hypereosinophilia, which can reach more than 50% of patients on long-term dobutamine according to other reports [15,16].

### 4.4. Other Strategies

Considering infectious complication associated with the intravenous catheter system, discontinuous infusion of Levosimendan could be an interesting therapy, due to its prolonged action, the possibility of combination with beta blockers, its renal-protective effect, and vasodilation [17].

Iterative Levosimendan injection may appear to be a less risky alternative. Unfortunately, this therapy has not yet shown any efficacy with studies that are currently neutral or negative [18,19].

### 4.5. Study Limitations

This study has certain limitations. Firstly, this is a retrospective observational study and not a randomized controlled trial.

The inotropic dependence of our patients and the minimum dose of dobutamine necessary were defined by clinical approach with echocardiography, not integrating the right heart catheterization in a systematic pattern.

There was no comparison group in our study, our patients were at the Intermacs 3, stable on dobutamine but were not weanable. They may not survive hospital discharge without inotropes. In this setting, it would be unethical to have a comparable non-inotropic group.

We did not assess the quality of life in our population. This parameter is of prime importance in palliative care patients.

## 5. Conclusions

Management of dobutamine-unweanable patients after cardiogenic shock may involve dobutamine at home to permit a home return. This strategy was associated with significant survival and few readmissions, and, if eligible, access to surgical strategies, such as heart transplantation. Simple biological markers at discharge could help identify severe patients for referral to palliative care and good responders, but further studies and especially randomized clinical trials are mandatory in this patient population.

## Figures and Tables

**Figure 1 jcm-10-02571-f001:**
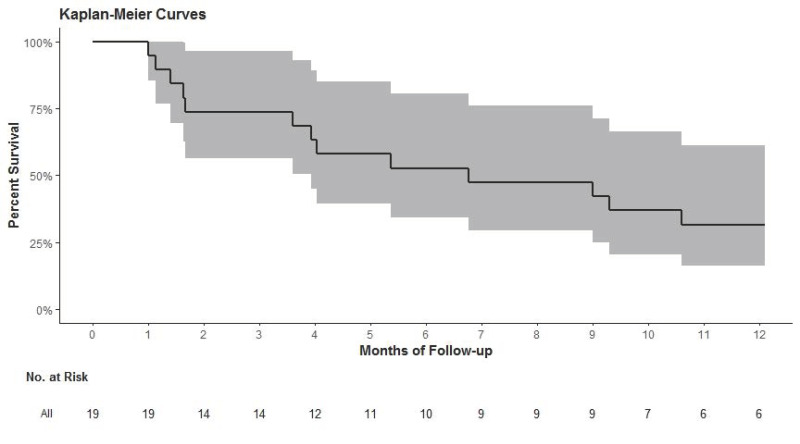
Event-free survival in the whole study population.

**Figure 2 jcm-10-02571-f002:**
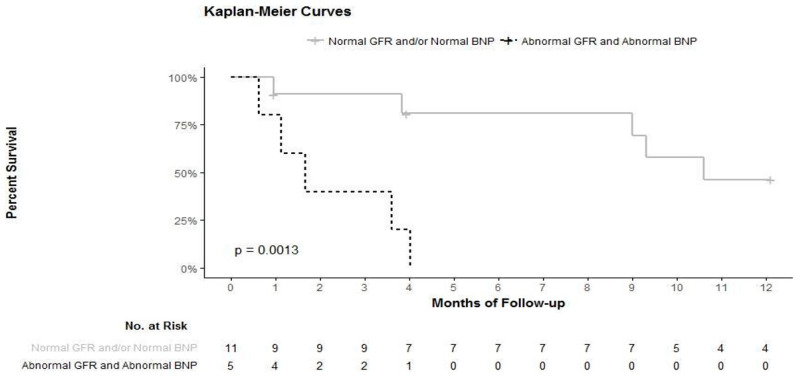
Event-free survival in the patient group with BNP > 1000 ng/L and GFR < 60 mL/min and all other patients from the cohort. An event was defined as all cause death.

**Figure 3 jcm-10-02571-f003:**
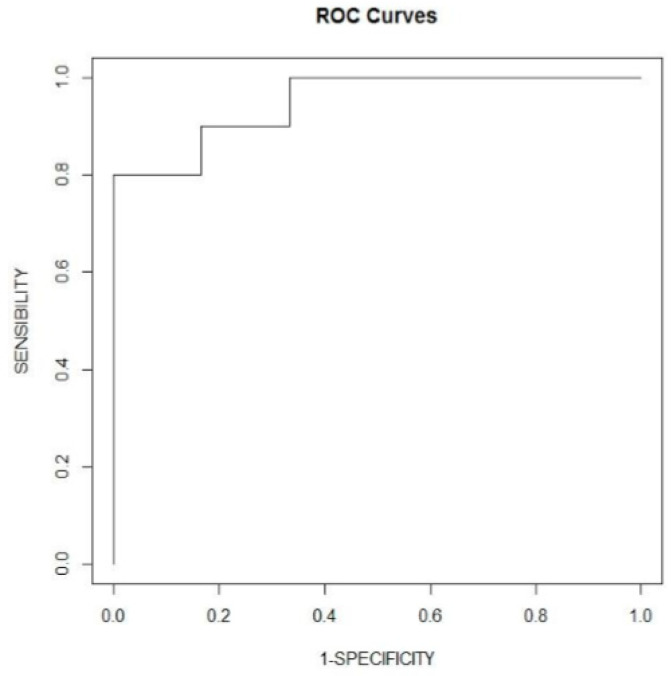
Receiver operating characteristic curves representing modelization of one-year mortality by GFR and BNP, adjusted for diabetes and FEVG.

**Table 1 jcm-10-02571-t001:** Baseline characteristics.

Parameter	Total Population (*n* = 19)
Demographics	
Male, *n* (%)	18 (95)
Age, years	65 ± 10
Octogenarian, *n* (%)	1 (5)
Ischemic etiology HF, *n* (%)	12 (63)
**Clinical characteristics**	
BMI, kg/m^2^	23.4 ± 3.5
Systolic BP, mmHg	88 ± 13
Diastolic BP, mmHg	58 ± 9
NYHA	3.2 ± 0.7
**Co-morbidities**	
PCI, *n* (%)	10 (53)
CABG, *n* (%)	3 (16)
Atrial fibrillation, *n* (%)	11 (58)
Valve surgery, *n* (%)	1 (5)
Diabetes, *n* (%)	9 (47)
COPD, *n* (%)	5 (26)
**Baseline therapy**	
ACE-I/ARB, *n* (%)	14 (74)
Beta-blockers, *n* (%)	9 (47)
MRA, *n* (%)	13 (68)
Loop diuretic, *n* (%)	14 (74)
Amiodarone, *n* (%)	9 (47)
ICD or CRT-D, *n* (%)	10 (53)
**Heart failure**	
LVEF (%)	21 ± 5
Destination therapy, *n* (%)	12 (63)
Bridge to treatment, *n* (%)	7 (37)
Dobutamine mean dose, γ/kg/min	2.6 ± 1.2

BMI: body mass index, PCI: percutaneous coronary intervention, CABG: coronary artery bypass surgery, COPD: chronic obstructive pulmonary disease, ACE-I: angiotensin converting enzyme inhibitor, ARB: angiotensin II receptor blockers, MRA: mineralocorticoids receptor antagonists, ICD: implantable cardioverter defibrillator, CRT-D: cardiac resynchronization therapy defibrillator, LVEF: left ventricular ejection fraction.

**Table 2 jcm-10-02571-t002:** Biomarkers at baseline and follow-up.

Laboratory Results	Admission Value	Discharge Value	Last Follow-Up	*p*-Value
Creatinine, mmol/L	153 (122.5–192.5)	120 (94–147)	138 (105–166)	0.012
Glomerular filtration rate, mL/min/1.74 m^2^	42 (33–55)	55 (42.5–75)	47 (38–67)	0.012
BNP, ng/L	1875 (1242–2705)	951 (353–1722)	1319 (618–2705)	0.197
Total bilirubin, µmol/L	17 (12–32)	16 (12–24)	15 (12–22)	0.07

BNP: brain natriuretic peptide GFR was calculated using the CKD-EPI formula.

**Table 3 jcm-10-02571-t003:** Total population and destination therapy population follow-up.

Follow-Up	Total Population, *n* = 9	Destination Therapy, *n* = 12
Heart transplantation	3 (16%)	0
PMVR	1 (5%)	0
LVAD	1 (5%)	0
Treatment duration, days	203 (79–434)	201 (94–347)
Alive at 30 days	18 (95%)	11 (92%)
Alive at 3 months	14 (74%)	9 (75%)
Alive at 6 months	10 (53%)	6 (50%)
Alive at 12 months	6 (32%)	3 (25%)
Time at home (since IHD start), days	163 ± 139	184 ± 139
Time at hospital (since IHD start), days	31 ± 23	35 ± 23

LVAD: left ventricular assist device; PMVR: percutaneous mitral valve repair.

**Table 4 jcm-10-02571-t004:** Univariate and multivariate Cox regression analyses on baseline characteristics associated with one-year mortality.

Baseline Characteristics	Univariate		Multivariate	
	HR, 95%CI	*p* Value	HR, 95%CI	*p* Value
BNP at discharge > 1000 ng/L	2.97 (0.78–11.39)	0.11	1.53 (0.22–10.69)	0.67
GFR at discharge < 60 mL/min	3.89 (1.17–12.95)	0.03	7.04 (1.32–37.48)	0.02

HR: hazard ratio; 95%CI: 95% confidence interval. Multivariate analyses were adjusted for diabetes and FEVG.

## Data Availability

The data will be made available upon all reasonable request with our sponsor’s authorization.

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
