# Peer review of "Outcome Predictors and Safety of Home Dobutamine Intravenous Infusion in End Stage Heart Failure Patients"

_jcm, 2021, doi:10.3390/jcm10122571_

Round 1

Reviewer 1 Report

In the absolute first paragraph of the ABSTRACT the authors inform the reader that "In patients with end-stage heart failure, ambulatory continuous intravenous dobutamine has been suggested to improve quality of life, reduce heart failure hospitalizations and associated cost-expenditures.". However, in the INTRODUCTION, the same claim is not supported by any evidence, and the reader learns instead that "intravenous home dobutamine infusion [is] not a routine procedure and there is little data on this type of management in the literature" (see references 3 and 4).

Indeed "home dobutamine" was suggested already in the 1990 by Collins et al. [J Heart Transplant. 1990;9(3 Pt 1):205-8] as bridge to transplant. 7 other PubMed-listed publications returns to the subject, but none of them pass the "clinical trial" filter on PubMed. There is simply no evidence-based medicine on this.

At the end of the CONCLUSION the reader is reminded that "If [more trials are]not done, then we will continue in the same series of empirical and subjective setting of therapies without evidence, with heterogenous results, potential safety issues and no proof of actual efficacy". It seems that the present manuscript - with an analysis of retrospective data from few patients - offers very little to evidence-based medicine in this therapy segment.

The author should change radically their narrative to make the manuscript useful to the clinical community.

Author Response

Dear editor and reviewers

First, we would like to thank you for reviewing our publication and for providing comments. We have integrated all remarks and addressed all the queries that we could..Please find attached files with separated answer and updated manuscript.

Reviewer 2 Report

Thanks for the opportunity of reviewing this interesting manuscript.

Jobbe-Duval et al report a single center experience of 19 patients who were discharged on home dobutamine between 2/2014 and 11/2019 with survival rate of 32% at 1 year, in that cohort there were 5 patients who had a catheter related adverse events.

Comments

-It would be important for the authors to provide information about the practice at their institution. How many patients are evaluated for advanced heart failure at their institution. How many LVADs are inserted as a bridge to transplant?

-As the authors pointed out in their discussion the survival of palliative inotropes is 33% at 1 year. I would try to avoid mixing the BTT and palliative care group.

In the palliative care group, the impact in quality-of-life measures would be more important than survival.

-What precluded the 7 patients in BTT group to get an LVAD? Right ventricular failure? Other?

-Can we provide information about baseline hemodynamics?

-Were all patients in BTT listed for transplantation?

-What was the total time on Dobutamine Therapy?

-Some programs use milrinone, any reason to avoid this drug?

-In table 1 mentions that 47% of the patients were on a beta-blocker. Using dobutamine in the context of beta-blockers is ineffective.  Can the authors provide clarification of this issue?

-It has been described that the Seattle HF model underestimates the risk of death in patients with advanced heart failure so I would not use it as a comparator.

-Would re-structure the discussion

A-Findings

B-Previous studies. Please review the following:

Assad-Kottner C, Chen D, Jahanyar J, Cordova F, Summers N, Loebe M, Merla R, Youker K, Torre-Amione G. The use of continuous milrinone therapy as bridge to transplant is safe in patients with short waiting times. J Card Fail. 2008 Dec;14(10):839-43. doi: 10.1016/j.cardfail.2008.08.004. Epub 2008 Nov 5. PMID: 19041047.

C-Other

D-Limitations

E-Conclusions.

Author Response

Dear editor and reviewers

First, we would like to thank you for reviewing our publication and for providing comments. We have integrated all remarks and addressed all the queries that we could..Please find attached files with separated answer.

Round 2

Reviewer 1 Report

the authors accepted my suggestions and corrected the manuscript accordingly

Reviewer 2 Report

The authors have updated the manuscript following the recommendations. No further suggestions/ comments